# Sensitivity Enhancement and Probiotic Detection of Microfluidic Chips Based on Terahertz Radiation Combined with Metamaterial Technology

**DOI:** 10.3390/mi13060904

**Published:** 2022-06-07

**Authors:** Yen-Shuo Lin, Shih-Ting Huang, Shen-Fu Steve Hsu, Kai-Yuan Tang, Ta-Jen Yen, Da-Jeng Yao

**Affiliations:** 1Institute of NanoEngineering and MicroSystems, National Tsing Hua University, Hsinchu 30013, Taiwan; eason0920200235@gmail.com; 2Department of Power Mechanical Engineering, National Tsing Hua University, Hsinchu 30013, Taiwan; doublu1208@gmail.com; 3ACE BIOTEK Co., Ltd., Hsinchu 30261, Taiwan; steve_hsu@acesolution.com.tw (S.-F.S.H.); david_tang@acesolution.com.tw (K.-Y.T.); 4Department of Material Science Engineering, National Tsing Hua University, Hsinchu 30013, Taiwan; tjyen@mx.nthu.edu.tw

**Keywords:** terahertz, metamaterials, microfluidics, sensitivity enhancement, probiotic

## Abstract

Terahertz (THz) radiation has attracted wide attention in recent years due to its non-destructive properties and ability to sense molecular structures. In applications combining terahertz radiation with metamaterial technology, the interaction between the terahertz radiation and the metamaterials causes resonance reactions; different analytes have different resonance performances in the frequency domain. In addition, a microfluidic system is able to provide low volume reagents for detection, reduce noise from the environment, and concentrate the sample on the detection area. Through simulation, a cruciform metamaterial pattern was designed; the proportion, periodicity, and width of the metamaterial were adjusted to improve the sensing capability of the chip. In the experiments, the sensing capabilities of Type A, B, and C chips were compared. The Type C chip had the most significant resonant effect; its maximum shift could be increased to 89 GHz. In the probiotic experiment, the cruciform chip could have a 0.72 GHz shift at a concentration of 0.025 mg/50 μL, confirming that terahertz radiation combined with a metamaterial microfluidic chip can perform low-concentration detection.

## 1. Introduction

Terahertz technology and its applications have developed rapidly in recent years. Terahertz (THz) radiation generally refers to the frequency band ranging from 0.1 to 10 THz, between the microwave and infrared regions of the electromagnetic spectrum [1]. Compared to electromagnetic waves in other ranges, terahertz radiation can penetrate most non-polar materials without destroying the material, so terahertz technology is deemed attractive for various applications, including biomedical imaging, packaged goods inspection, food inspection, and water pollution detection [2]. In addition, terahertz radiation may be easily absorbed by polar materials, such as water molecules and ammonia molecules, which are characterized by both vibrational and rotational transitions occurring in the terahertz frequency band [3]. These molecular structures have different resonance modes and changes in the refractive index and absorbance, as well as exhibiting different spectral characteristics; thus, terahertz radiation can be used to identify various biochemical structures. Terahertz spectroscopy can provide non-contact, non-destructive, label-free optical sensing for chemical and biological research, thereby achieving the purpose of rapid screening [4].

By designing material patterns with geometric shapes, these materials are artificially arranged into a series of subwavelength structures that exhibit specific properties of relative permittivity and magnetic permeability; these are called metamaterials [5,6]. When metamaterials interact with electromagnetic waves, periodically arranged metamaterial elements can become effective electromagnetic scattering media at wavelengths far exceeding the distance between elements [7]. Once a metamaterial interacts with an electromagnetic wave, it forms frequency resonance with a high-quality factor, a physical phenomenon that produces asymmetric linear resonance [8]. In the frequency spectrum, the signal can be observed to drop sharply and then rise, like a pointer in the frequency domain. Therefore, the combination of terahertz radiation and metamaterials can increase the sensitivity of terahertz radiation and enable label-free detection, making it valuable for future biomedical applications.

As the development of microelectromechanical systems (MEMS) technology has become increasingly mature, the size of the lithography process has been reduced to the micrometer or even nanometer scale, leading to the development of microfluidic technology. Microfluidics is a powerful technology that has been widely used to analyze biological particles in small quantities of liquids; integrate solution mixing, dilution, material separation, extraction, and detection into a chip; and give rise to lab-on-a-chip (LOC) [9]. Based on the concepts introduced above, this study integrates THz, metamaterial, and microfluidic system technologies, and conducts a series of experiments to confirm the ability of THz chips to detect probiotics and low-concentration differential samples.

## 2. Microfluidic System, Metamaterials and Experimental Setup

### 2.1. Microfluidic System and Metamaterial

Since polydimethylsiloxane (PDMS) is a highly biocompatible polymer with the advantages of a moderate cost and simple fabrication, it is a common microfluidic chip material applied in the biomedical field. The design of the microfluidic chip is shown in Figure 1a. The inlet and outlet radius of the microfluidic channel is 2.5 mm, the width of the channel is 1 mm, the radius of the detection area is 4 mm, and the surrounding cross pattern is the alignment mark. As shown in Figure 1b, considering the strong absorption property of liquid for terahertz radiation in the detection area, the height of the flow channel was designed to be 40 μm.

When terahertz radiation is combined with artificially engineered metamaterials, the quadrupole resonance mode of an X-shaped plasmonic sensor (XPS) can generate quadrupole resonance with reflection cancellation and result in high absorption, a high quality factor, and high sensitivity [10]. As shown in Figure 1c, in the design of the XPS unit, the length was 350 μm, the width was 100 μm, and the angle of 60 degrees was used. The unit metamaterials were arranged into a metamaterial array with a period length (periodicity) of 450 μm, which was designed in the detection area of the chip to resonate with THz frequencies.

The simulated resonance position of the chip containing the metamaterial array was at 0.295 THz, as shown in Figure 1d. The spectrum presents a narrow bandwidth resonant dip, which means a high FOM and quality factor [11]. This unique resonance dip was used as an indicator for analysis by observing the X- and Y-axes’ resonance shift and variation in the resonant dip.

### 2.2. Chip Fabrication and Experimental Setup

The microfluidic chip was fabricated using photolithography. It can be divided into two layers: the bottom layer as the metamaterial, and the upper layer as the microfluidic system.

For the bottom layer, the wafer needed to be cleaned using standard processes, and the HMDS was evaporated for 5 min to enhance photoresist adhesion. Next, a positive photoresist (AZ5214) (SONO TEK Corporation, New York, NY, USA)was spin-coated at 3000 rpm for 30 s and soft-baked at 100 °C for 1 min. After that, exposure (energy density of 60 mJ/cm^2^) and development processes were carried out. An adhesion layer of 20 nm thickness and gold of 200 nm were then deposited, and, finally, the lift-off process removed excess material other than the metamaterial.

For the upper layer, first, the mold of the microfluidic system was made. The negative photoresist SU8 was directly coated on the substrate at a pre-spin speed of 500 rpm for 10 s and then spun at 3500 rpm for 30 s. Next, the soft bake started at 75 °C and was increased by 5 °C every 3 min until 95 °C was reached; this temperature was maintained for 15 min. After that, the exposure (energy density of 250 mJ/cm^2^) and development processes were carried. Finally, a hard bake was undertaken for 10 min to enhance the SU8 structure. After completing the SU8 mold of the microfluidic system, polydimethylsiloxane (PDMS) (agent A/agent B = 10:1) was poured into the mold of the microfluidic system and heated at 80 °C for 40 min to obtain the 2 mm-thick upper layer of the microfluidic system.

After the bottom and upper layers were made, the binding energy was used to assemble them. To achieve binding energy, hydrophobic properties must be transformed into hydrophilic at the interface through surface modification employing an oxygen plasma. The final product was completed by oxygen plasma binding and baking at 65 °C for 5 min. Figure 2a–d show the fabrication process, including the upper and lower layers and the binding technique.

When the THz radiation penetrated the chip vertically and the analyte existed on the surface of the metamaterial, the metamaterial environment had a dielectric constant difference, and the signal received by the detector changed. The experimental setup is shown in Figure 2e. Finally, the data were analyzed by observing the horizontal and vertical changes in the signal in the frequency domain.

There were two THz instruments used in the experiment: the TeraPulse System (TeraPulse 4000) and the VDI System. As shown in Figure 2f, the TeraPulse System uses THz time-domain spectroscopy (THz-TDS), and the generation of terahertz pulsed radiation is based on a photoconductive switch. Its resolution is 7.09 GHz, and the operating regime is from 100 GHz to 4000 GHz. The TeraPulse System is a closed measuring instrument. After the chip is fixed on the platform in the chamber, the instrument extracts vapor and injects dry air to reduce the effect of vapor on the absorption of terahertz waves. Another instrument is the VDI system, in which the wave source is the continuous wave generated by an RF electron beam, and the detector receives the signal in the frequency domain. It has a narrow regime of 360 GHz to 500 GHz and a high resolution of 0.36 GHz, as shown in Figure 2g.

## 3. Simulation

### 3.1. XPS Dimension Comparison

The XPS pattern had three geometrical designs on the quartz substrate, named Type A, Type B, and Type C, as shown in Figure 3a. The lengths of the XPS metamaterials were 350 µm, 240 µm, and 220 µm; the widths were 100 µm, 80 µm, and 70 µm; and the periodicities were 450 µm, 380 µm, and 300 µm. The simulation results are shown in Figure 3b. The resonance positions of the Type A, Type B, and Type C chips were 0.295 THz, 0.389 THz, and 0.451 THz, and the resonance positions were gradually blue-shifted. In terms of resonance strength, the resonance strengths of the Type A, Type B, and Type C chips were 21.8, 23.2, 26.9 (units); the resonance strength gradually increased.

In the *X*-axis resonance position section, Type A had the lowest frequency resonance position, while Type C had the highest frequency resonance position. The reason is that Type A was the largest in geometric size and had a lower natural resonance frequency, while Type C was the smallest in geometric size and had a higher natural resonance frequency. In the *Y*-axis resonance strength section, Type A had the weakest resonance strength, while Type C had the most obvious resonance strength. The reason is that Type A had the longest periodicity and fewer metamaterials per unit area to produce a resonance effect, and the resonance strength was weak; on the contrary, Type C had the shortest periodicity and had the most metamaterials per unit area to produce resonance, and the resonance effect was the most significant. Therefore, the subsequent simulations used Type C for observation and analysis.

### 3.2. Adjustment of Sensitivity and Resonance Position

By adjusting the angle and geometric parameter of the XPS metamaterial, the sensitivity of the chip can be improved and the resonance position can be adjusted. In the definition of sensitivity, as shown in the formula, S is the sensitivity, Δ*f* is the frequency shift, and Δ*n* is the refractive index change. It can be determined from the formula that the sensitivity, *S*, is the ratio of the frequency shift to the refractive index change [12].
$$S=\frac{\Delta f}{\Delta n}$$
where *S* is sensitivity, Δ*f* is frequency, and Δ*n* is change of Refractive index.

The originally designed XPS pattern was 60 degrees; the angle was reduced to 30 degrees and increased to 90 degrees, as shown in Figure 4a. The sensitivities of the three XPSs and the ratio of the frequency change to the refractive index change, as shown by the three slopes in Figure 4a, were observed. The 30 degrees (XPS) had a sensitivity of 51.8 GHz/RIU; the 60 degrees (XPS) had a sensitivity of 55.5 GHz/RIU; and the 90 degrees (XPS) had a sensitivity of 57.4 GHz/RIU.

Then, the XPS 90 degrees, Cruciform 90 degrees, and CSA 90 degrees were compared. The difference is that the Cruciform 90 degrees is a cross shape with smooth corners, while the CSA 90 degrees is a cross shape with vertical corners, as shown in Figure 4b [12]. The XPS 90 degrees had a sensitivity of 57.4 GHz/RIU; the CSA 90 degrees had a sensitivity of 59.2 GHz/RIU; and the Cruciform 90 degrees had a sensitivity of 61.1 GHz/RIU. In conclusion, the Cruciform 90 degrees could most effectively improve the sensing ability of the chip and had the best sensitivity, so the following simulations used the Cruciform 90 degrees for observation and analysis.

In the section of resonance position adjustment, the proportion, periodicity, and width of the unit Cruciform metamaterial were adjusted. First, by increasing the proportion of the unit Cruciform metamaterial, the redshift effect could be produced. As shown in Figure 5a, the proportion of the Cruciform metamaterial was gradually enlarged from 70 µm to 75 µm, 80 µm, 85 µm, and 90 µm, and the geometric ratio was changed from 7.1%, 14.3%, 21.4%, and 28.6%, with 34, 67, 95, and 118 GHz redshifts, respectively. Second, by increasing the periodicity of the unit Cruciform metamaterial, the redshift effect could be produced. As shown in Figure 5b, the periodicity of the Cruciform metamaterial was gradually increased from 300 µm to 320 µm, 340 µm, 360 µm, and 385 µm, and the geometric ratio was changed from 6.7%, 13.3%, 20.0%, and 28.3%, with 16, 32, 46, and 62 GHz redshifts, respectively. Third, by reducing the width of the unit Cruciform metamaterial, the redshift effect could be produced. As shown in Figure 5c, the width of the Cruciform metamaterial was gradually reduced from 70 µm to 65 µm, 60 µm, 55 µm, and 50 µm, and the geometric ratio was changed from 7.1%, 14.3%, 21.4%, and 28.6%, with 9, 18, 26, and 35 GHz redshifts, respectively.

The redshift shift value caused by adjusting the proportion, periodicity, and width of the Cruciform metamaterial was divided by the geometric ratio change to determine which method had the most significant redshift effect. As shown in Figure 5d, the first method—increasing the proportion—had the most significant redshift effect, with 4.1 GHz/% change; the second method—increasing the periodicity—had a normal redshift effect, with 2.2 GHz/% change; and the third method—reducing the width—had the least significant redshift effect, with 1.2 GHz/% change.

By observing the simulation results in this section, it was determined that the sensitivity of the Cruciform metamaterial was better than that of XPS, and the resonance position could be determined by changing the proportion, periodicity, and width of the Cruciform metamaterial. Therefore, a new metamaterial pattern was designed; the metamaterial was designed into a Cruciform shape, the periodicity of the metamaterial was reduced to 270 µm, and the width was reduced to 40 µm, as shown in Figure 6. In the next section, we use probiotic experiments to compare the sensitivity difference between the newly designed Cruciform metamaterial and the original XPS in order to confirm that the Cruciform metamaterial has better sensitivity.

### 3.3. The Influence of Dielectric Constant and Absorption Coefficient on the Position of X- and Y-Axes

The Cruciform metamaterial in Figure 6 was used to simulate the effect of changing the dielectric constant and absorption coefficient on the resonance position of the X- and Y-axes.

The dielectric constants of the analytes were 1, 1.5, 2.5, 3.5, and 4.5 (refractive index positions of 1.00, 1.22, 1.58, 1.87, and 2.12), and the resonance curves are shown as five blue resonance curves in Figure 7a. The *X*-axis resonance positions were 455, 430, 390, 360, and 335 GHz, and the resonance shifts were 25, 65, 95, and 120 GHz. As the refractive index increased, the resonance position shifted towards a lower frequency. Taking the refractive index as the *X*-axis and the resonance shift as the *Y*-axis, a positive correlation line with a correlation coefficient R^2^ = 0.9992 could be obtained, as shown in Figure 7b.

The absorption coefficients of analytes were 50, 100, 150, and 200 (cm^−1^), and the resonance curves are shown as four yellow resonance curves in Figure 7c. The *X*-axis resonance positions were all located at 455GHz, and the *Y*-axis resonance positions were −21.59, −17.98, −14.95, and −12.05 (S21dB). As the absorption coefficient increased, the *X*-axis resonance position did not change, while the *Y*-axis electric field gradually decreased. Taking the absorption coefficient as the *X*-axis and the electric field as the *Y*-axis, a positive correlation line with a correlation coefficient R^2^ = 0.9973 could be obtained, as shown in Figure 7d.

From the above simulation of changing the dielectric constant and absorption coefficient sequentially, it was observed that the change in the dielectric constant affected the different resonance positions of the *X*-axis, and the change in the absorption constant affected the different electric field of the *Y*-axis, as shown in Figure 7e.

## 4. Results

### 4.1. XPS Dimension Comparison

In Section 3.1, it was observed that, among the Type A, B, and C chips, Type A has the weakest resonant strength, and Type C has the most obvious resonant strength. In the experiment, IPA solutions of different concentrations (0%, 25%, 50%, 75%, and 100% IPA) were distinguished using the Type A, B, and C chips. As shown in Figure 8, when the liquid was injected into the chip, we could see that different concentrations of IPA had different shift values due to the different dielectric properties of the liquid (n_IPA_ = 1.51; n_Water_ = 2.2) [13].

As shown in Figure 8a, in the Type A chip, the resonance frequency was 0.292 THz when the microchannel was full of air. When injecting IPA from high to low concentrations in the microchannel, the shift increased from 0.035 THz to 0.044 THz, 0.053 THz, 0.061 THz, and 0.068 THz. In the Type B chip, the resonance frequency was 0.386 THz when the microchannel was full of air. When injecting IPA from high to low concentrations in the microchannel, the shift increased from 0.037 THz to 0.049 THz, 0.061 THz, 0.070 THz, and 0.081 THz. In the Type C chip, the resonance frequency was 0.449 THz when the microchannel was full of air. When injecting IPA from high to low concentration in the microchannel, the shift increased from 0.039 THz to 0.055 THz, 0.067 THz, 0.079 THz, and 0.089 THz.

A comparison of the shifts in the Type A, B, and C chips is shown in Figure 8b. In Section 3.1, in terms of resonant strength, Type A had the weakest resonant strength, while Type C had the most pronounced resonant strength. A larger resonant strength means that the resonance effect is more significant, so Type C could also cause the largest resonance shift in the experiment and had higher sensitivity.

### 4.2. Probiotic Experiment

Through the probiotic experiment, the sensitivity difference between the newly designed Cruciform metamaterial and the original XPS (Type C) in Section 3.2 was compared, in order to confirm that the Cruciform metamaterial has better sensitivity. We deployed several concentrations of Lactobacillus aqueous solution (0.025 mg, 0.05 mg, 0.125 mg, 0.25 mg, and 0.5 mg per 50 μL), dropped them on the surface of the XPS and Cruciform chips without the microchannel, and performed experiments using the VDI System.

The measurement results of the Type C metamaterial chip are shown in Figure 9a. the initial resonance position is at 435.92 GHz, and the resonance positions from low to high concentration are 435.56 GHz, 434.8 GHz, 433.04 GHz, 431.2 GHz, and 425.4 GHz. The measurement results of the Cruciform metamaterial chip are shown in Figure 9b. Thehe initial resonance position is 438.08 GHz, and the resonance positions from low to high concentration are 437.36 GHz, 436.24 GHz, 433.72 GHz, 430.12 GHz, and 421.44 GHz.

A comparison of the shift in the Type C and Cruciform chips is shown in Figure 9c. For the shift in the Type C metamaterial chip, the concentration from low to high is 0.36 GHz, 1.12 GHz, 2.88 GHz, 4.72 GHz, and 10.52 GHz; for the shift in the Cruciform metamaterial chip, the concentration from low to high is 0.72 GHz, 1.84 GHz, 4.36 GHz, 7.96 GHz, and 16.64 GHz. It can be seen that the shift in the Cruciform metamaterial is greater than that in the Type C metamaterial at the five different concentrations. Moreover, the probiotic concentration slope increased from 20.9 for the Type C chip to 33.0 for the Cruciform metamaterial chip. The above results show that Cruciform metamaterial chips have better sensing capabilities than those of Type C metamaterial chips.

### 4.3. Mixed Solution with Small Concentration Difference

In order to test the Cruciform metamaterial with higher sensitivity, analytes with small concentration differences were used to observe the sensing ability of the chip. In the experiment using the mixed solution with a small concentration difference, the concentration was between 95% and 100%. The following two kinds of mixed solution experiments were carried out: 1. isopropanol (IPA) + ethanol; 2. methanol + acetone mixed solution.

In the experiment using the mixed solution of isopropanol (IPA) + ethanol, as shown in Figure 10a, the *X*-axis resonance positions of 95%~100% IPA were all located at 0.396 THz. At the *Y*-axis position, it could be observed that with the decrease in the IPA concentration, the electric field weakened, the *Y*-axis resonance position moved from −5.309 to −4.975 (Units), and the *Y*-axis position moved upward. In the experimental using the mixed solution of methanol + acetone, as shown in Figure 10b, the *X*-axis resonance positions of 95%~100% methanol were all located at 0.312 THz. At the *Y*-axis position, it could be observed that with the decrease in the methanol concentration, the electric field weakened, the *Y*-axis resonance position moved from −2.561 to −2.345 (Units), and the *Y*-axis position moved upward.

At low concentration differences, different analytes could be identified by the *Y*-axis. The reason is that when the concentration of IPA/methanol decreased, the absorption coefficient of the solution increased, resulting in the weakening of the resonance effect, as shown in Figure 10c [14].

## 5. Conclusions

In the analysis of the XPS metamaterial, the resonance strength and shift in Type A, B, and C chips were compared. In terms of resonance strength, the simulation and experimental results show that Type A had the weakest resonance electric field, while Type C had the most obvious resonance electric field, which could also cause the largest resonance shift; the maximum shift could be increased to 89 GHz. Through simulation, it was determined that the Cruciform metamaterial had better sensitivity. Additionally, it was observed that the change in the dielectric constant affected the different resonance positions of the *X*-axis, and the change in the absorption constant affected the different electric fields of the *Y*-axis. In probiotic experiments, Type C and Cruciform chips had shifts of 0.36 and 0.72 GHz at 0.025 mg/50 μL, respectively. In the mixed solution experiment with a low concentration difference, different analytes were identified by the difference in the *Y*-axis of the resonance position. This is the result of successful ultra-fast and low-concentration detection through terahertz radiation combined with metamaterial microfluidic chips.

## Figures and Tables

**Figure 1 micromachines-13-00904-f001:**
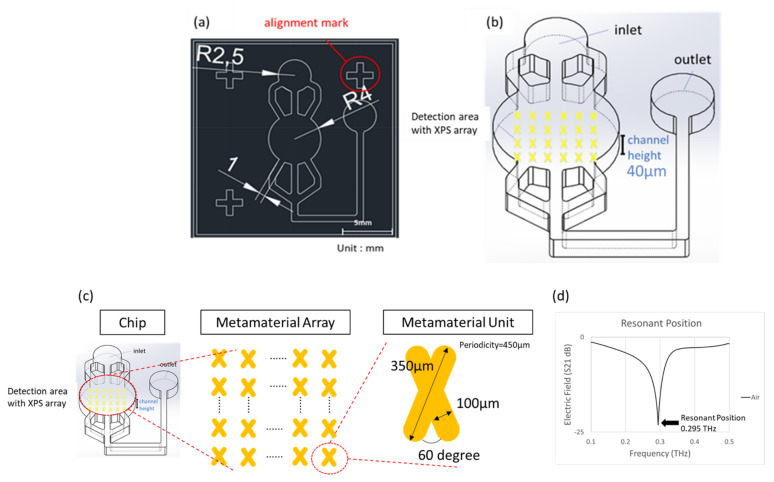
(**a**) The design of the microfluidic system. (**b**) Perspective view of the microfluidic system. (**c**) The design of the XPS metamaterial. (**d**) Simulation result shows a resonant dip at frequency of 0.295 THz.

**Figure 2 micromachines-13-00904-f002:**
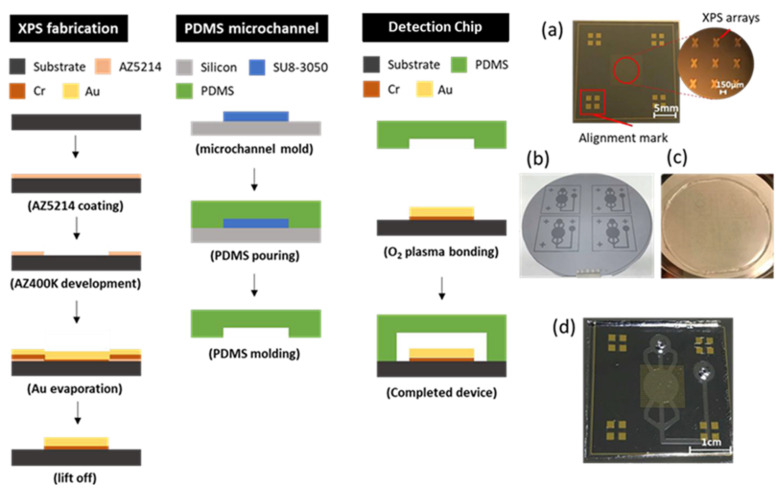
Fabrication process flow and (**a**) metamaterial; (**b**) SU8 mold; (**c**) PDMS microfluidic layer; (**d**) finished product (detection chip). (**e**) Schematic diagram of experimental setup; (**f**) TeraPulse System structure diagram; (**g**) VDI System structure diagram.

**Figure 3 micromachines-13-00904-f003:**
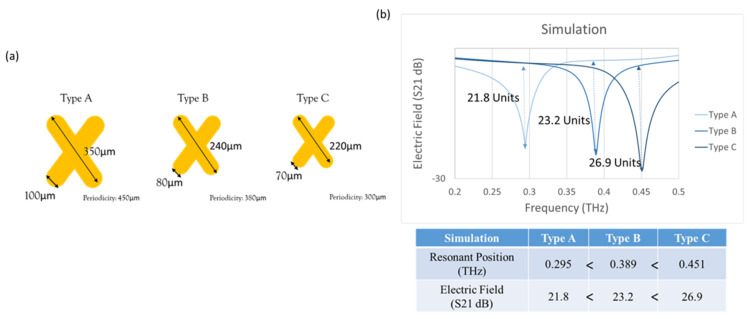
(**a**) Schematic diagram. (**b**) Simulated resonance position of three XPS pattern designs.

**Figure 4 micromachines-13-00904-f004:**
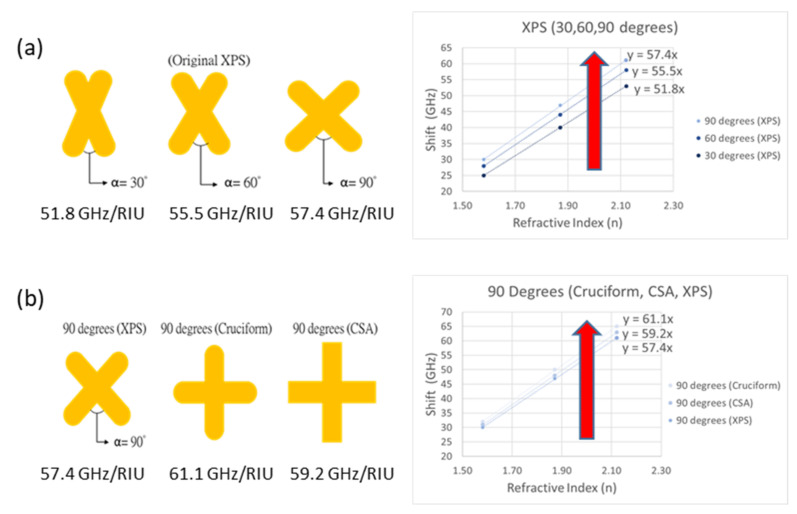
(**a**) Sensitivity of XPS (30, 60, 90 degrees). (**b**) Sensitivity of 90 degrees (XPS, Cruciform, CSA).

**Figure 5 micromachines-13-00904-f005:**
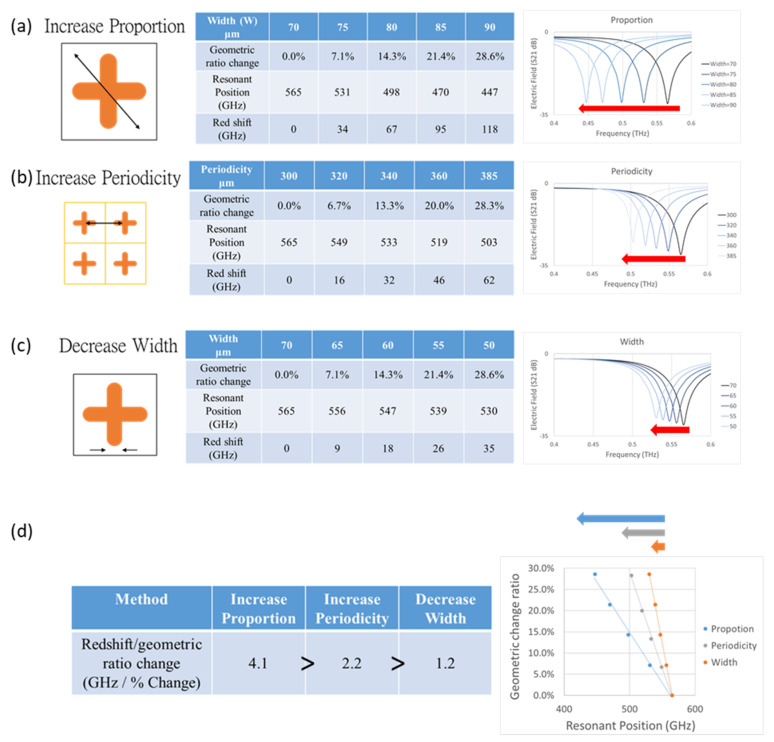
Adjustments to the resonance position by (**a**) increasing proportion, (**b**) increasing periodicity, and (**c**) decreasing width. (**d**) Comparison of redshift/geometric ratio change.

**Figure 6 micromachines-13-00904-f006:**
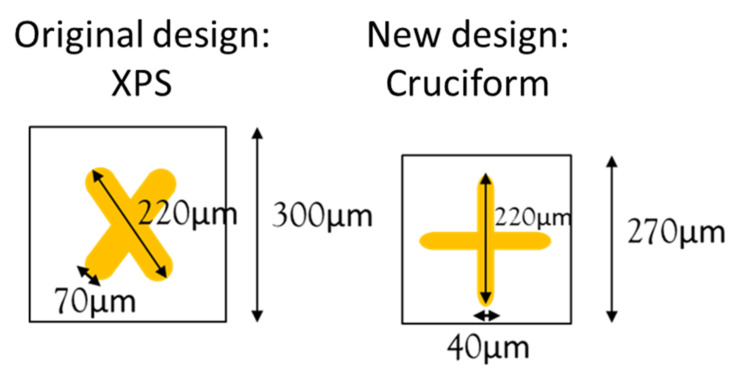
Schematic diagram of original design (XPS) and new design (Cruciform).

**Figure 7 micromachines-13-00904-f007:**
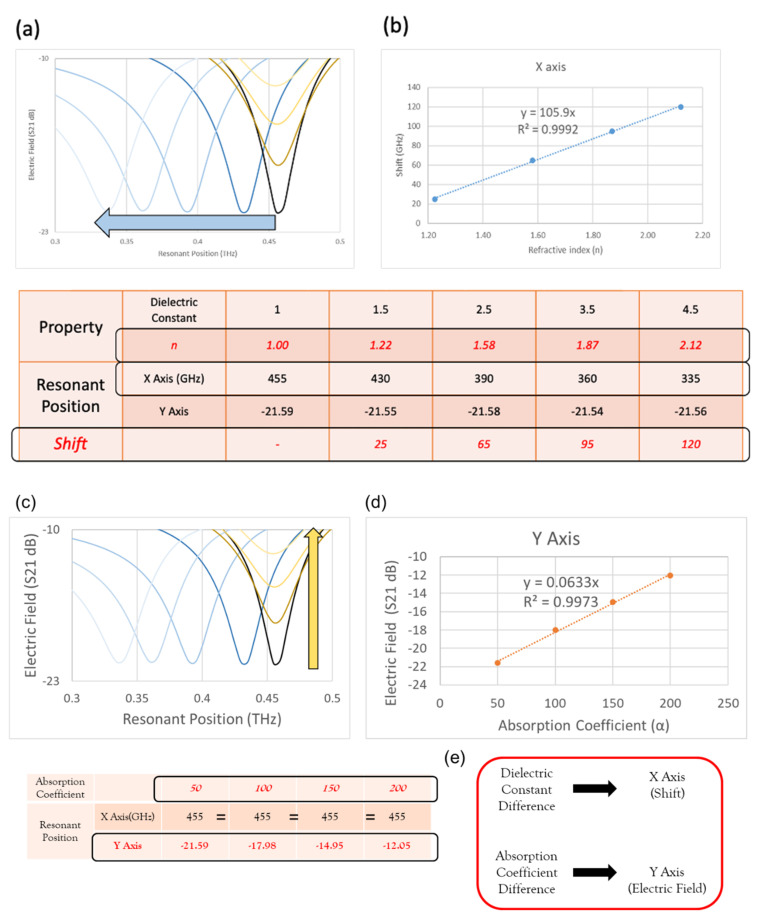
(**a**) Resonance curves of analytes with dielectric constants of 1, 1.5, 2.5, 3.5, and 4.5. (**b**) The relationship between the refractive index and shifts. (**c**) Resonance curves of analytes with absorption coefficients of 50, 100, 150, and 200. (**d**) The relationship between absorption coefficients and the electric field. (**e**) The influence of the dielectric constant and absorption coefficient on the position of the X- and Y-axes.

**Figure 8 micromachines-13-00904-f008:**
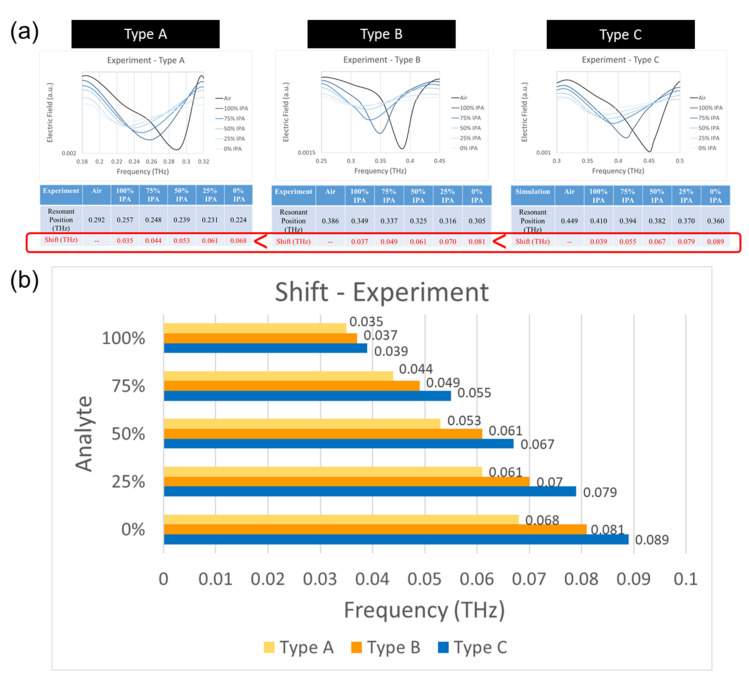
(**a**) Experimental results of different concentrations of IPA in Type A, B, and C chips. (**b**) Shift comparison chart for Type A, B, and C chips.

**Figure 9 micromachines-13-00904-f009:**
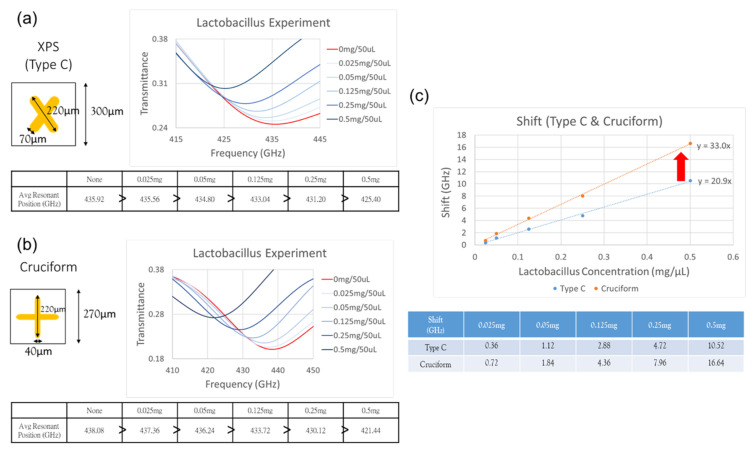
Probiotic measurement results of (**a**) Type C chip and (**b**) Cruciform chip. (**c**) Shift comparison chart of Type C and Cruciform chips.

**Figure 10 micromachines-13-00904-f010:**
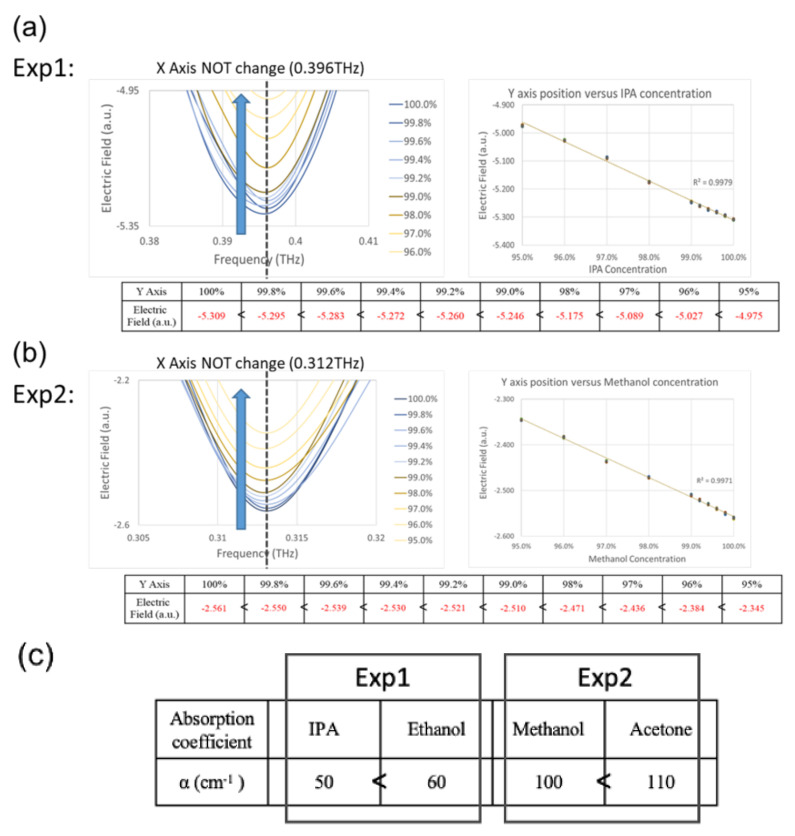
Experimental results of mixed solution of (**a**) IPA+ ethanol and (**b**) methanol + acetone. (**c**) Absorption coefficients of IPA, ethanol, methanol, and acetone.

## Data Availability

The data presented in this study are available on request from the corresponding author.

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
