# Peer review of "Sensitivity Enhancement and Probiotic Detection of Microfluidic Chips Based on Terahertz Radiation Combined with Metamaterial Technology"

_micromachines, 2022, doi:10.3390/mi13060904_

Round 1

Reviewer 1 Report

Dear Authors,

I accept the article after revision.

1) The spell check is required! Also in many places there is lack of the spaces, for instance between the numbers and units (in one place is 89 GHz with the space between, in the other 0.72GHz without); for instance line 21, 68, 98, 189, 211, 212, 258, 261, 303, 307, 327 etc. Also the spaces should be before the brackets with the references (for instance not "...spectrum[1]." but "spectrum [1]." - line 29). 

2) The presentation of results should be improved. Some graphs are not clear, the letters are too small (Figure 1(d) - the words describing the axis are almost non visible, similar with the Figure 3(b), Figure 4, Figure 5, Figures 8-10  - the graphs are of law quality and many information not visible. I reccomend to improve the pictures and their quality. Also consider to show the X metamaterial designs with proper proportions so the differences will be more visible.

3) The references are not prepared according to journal requirements. Check the style guide of the journal, please and correct the article according to it. The text and the pictures should be formatted in better way. 

4) Improve the introduction part with more latest literature references, please.

4) μL is not SI unit (for instance line 21) - please check with Editor if you can use this unit.

With regards,

your Reviewer

Reviewer 2 Report

Recommendation: major revision

I have thoroughly reviewed the manuscript entitled "Sensitivity Enhancement and Probiotic Detection of Microfluidic Chips Based on Terahertz Combined with Metamaterial Technology". I think that the current manuscript is an incomplete manuscript for publication because the formation of this manuscript is inadequate and detailed explanation is not sufficiently provided. My feeling is that the authors need to study a few examples of microfluidics-based advanced system reported in “high impact journals” before to resubmit the article. Taking into account the quality of work and scope of the journal, I would recommend the major revision according to the following comments.

Major comments:

# 1. This manuscript demonstrated the development of microfluidic system based on terahertz. Experimental results of this work are interesting, although the innovative points of this paper did not demonstrated clearly in the introduction part, experimental section, and results and discussions is not clearly presented.

# 2. The authors should improve their paper-writing with correct grammar and scientific scope.

# 3. Authors should be checked the format of this journal. Please read carefully the author guideline in this journal. Current manuscript is not acceptable.

Minor comments:

# 1. Figure 1: alignment mark is not important for conducting this experiment. The author should be modified the main figures for the clear understanding of the readers.

# 2. The author should have modified main figures in the whole of manuscript. Please manually handle the figures for the alignment of image materials (i.e., graph, image, table, etc.).

# 3. Although the research topic raised by author is interesting and crucial in microfluidics fields, I am afraid that the current version of the manuscript is incomplete (i.e., figures, tables, and graphs). This makes it difficult for the readers to fully appreciate the work presented.

Round 2

Reviewer 2 Report

The manuscript was corrected to the sufficient level for Micromachines.